# Measuring Disability Inclusion: Feasibility of Using Existing Multidimensional Poverty Data in South Africa

**DOI:** 10.3390/ijerph18094431

**Published:** 2021-04-22

**Authors:** Marguerite Schneider, Helen Suich

**Affiliations:** 1Department of Psychiatry and Mental Health, Alan J Flisher Centre for Public Mental Health, University of Cape Town, Cape Town 7700, South Africa; 2Crawford School of Public Policy, Australian National University, Canberra 2601, Australia; helen@helensuich.net

**Keywords:** measurement, multidimensional poverty, disability inclusion, Washington Group on Disability Statistics, South Africa

## Abstract

This paper presents a framework for measuring disability inclusion in order to examine the associations between disability severity and levels of inclusion, provides an example of its operationalization, and assesses the feasibility of using an existing dataset to measure disability inclusion using this framework. Inclusion here refers to the extent to which people with disabilities are accepted and recognized as individuals with authority, enjoy personal relationships, participate in recreation and social activities, have appropriate living conditions, are able to make productive contributions, and have required formal and informal support. Indicators for the operationalization were drawn from the Individual Deprivation Measure South Africa country study and were mapped on to the domains of inclusion (where relevant), and the Washington Group Short Set of questions were used to determine disability status (no, mild, or moderate/severe disability). The analysis indicates that individuals with disabilities experience generally worse outcomes and a comparative lack of inclusion compared to individuals without disabilities, and broadly that those with moderate or severe disabilities experience worse outcomes than those with mild disabilities. This analysis also provides insight into the limitations of using existing datasets for different purposes from their original design.

## 1. Introduction

Social inclusion is a standard and value that underpins all disability policies and programs to reverse the marginalization and related disadvantage of people with disabilities. Social inclusion has been described in the context of disadvantage generally as being complex and multidimensional [1], though descriptions range from a narrow conceptualization of simple economic inclusion [2] through to a broader conceptualization of social, cultural, economic, and political inclusion where conditions are such that individuals and groups are able to take part in society [3]. This literature on social inclusion is extensive but does not seem to effectively address disability as a factor that leads to exclusion. Furthermore, the disability inclusion literature is generally focused on small-scale studies, which are often qualitative in nature, generating a gap in how we can measure inclusion at scale—for example, to provide national-level statistics.

The themes in the broad social inclusion literature are, nevertheless, reflected to some degree in the literature on inclusion of people with disabilities. Hall sets out three key elements of social inclusion for people with disabilities, on the basis of her meta-analysis of the use of social inclusion in qualitative studies on disability. These are involvement in activities, maintaining reciprocal relationships, and a sense of belonging [4]. Ikäheimo further identifies institutional status and interpersonal status as separate features of personhood. Institutional status refers to the obligations of states towards their citizens, with respect to enabling the right to life, health, and education, and being counted as a human being. One clear example is whether a person has been issued with a formal identity document or birth certificate, which can enable them to access public services necessary to accessing equal opportunities. Interpersonal status refers to ‘being seen as a person by others’ ([5] p. 79), through being seen to have authority and a recognized claim to happiness, and to be contributing something worthy. Being seen as person ensures a person with disabilities is included into the ‘us’ by society and not dismissed as ‘them’—a group apart from mainstream society. 

It is well documented that people with disabilities are often excluded from educational and employment opportunities [6], and that accessible transport and public buildings, and availability of assistive devices and person assistance play key roles in facilitating inclusion [7]. The UN’s flagship ‘Disability and Development Report’ [6] highlights the common understanding that stigma and discrimination are key factors in hindering inclusion for people with disabilities. What is less extensively documented is the extent to which other aspects of inclusion are experienced, such as being respected and valued, having sufficient access to food and accommodation, and having a voice through being able to vote freely.

Hall identified six themes relating to inclusion from her review of 15 primary qualitative studies: (i) being accepted, (ii) having relationships, and (iii) involvement in activities supported by (iv) adequate living accommodations (e.g., including accessible dwellings, access to water, energy, and sanitation) and (v) support systems (e.g., formal service provider support and less formal support from family, friends, and community) and reflected in (vi) employment [4]. The ability to make choices and being part of and being seen in a range of social contexts (spaces and events including in employment and education) was also noted as being key to a person with disabilities feeling included [8,9]. Being given opportunities to reciprocate, being valued, and being expected to contribute to, for example, community events, foster agency and recognition and contribute to making a person feel real [9,10]. Having a voice involves being able to vote, being part of decision making at a personal level and within local activities, and being heard [8,10,11]. While a number of the studies in Hall’s review were conducted in high-income countries, these concepts remain relevant for any context, although the way they are realized may differ by cultural and geographical context.

In their review of social inclusion and people with disabilities in West Africa, Jolley et al. conclude that measures of disability and social inclusion need to be developed and adopted to allow for a more coordinated monitoring of social inclusion [12]. Collecting data through surveys is an important component of monitoring of inclusion as it provides an indication of the extent of inclusion across the broad target population and can record positive or negative changes over time. However, these data are only valid if measurement accurately reflects the notion of inclusion.

On the basis of this review of the disability studies literature, we propose a framework of the key domains of inclusion that should be included in large-scale studies measuring levels of inclusion of people with disabilities. We then provide an example of the operationalization of elements of this framework examining associations between disability severity and levels of inclusion using existing data from the Individual Deprivation Measure (IDM) South Africa Country Study. Given the financial and time costs associated with large data collection exercises, it is useful to identify existing datasets that allow us to identify indicators of inclusion to monitor these at the population level, for example for reporting on the Sustainable Development Goals, and we thus use the IDM South Africa data to assess the feasibility of using existing datasets to measure inclusion.

By using existing data to measure inclusion, we can gain useful insights not only on the associations between inclusion and disability severity in South Africa, but also on issues around utilizing data originally designed to measure something different (in this case, levels of individual deprivation). Poverty is broadly understood as an inability to achieve a socially acceptable standard of living across multiple dimensions. The IDM was designed to measure individual-level deprivation, capable of revealing gender disparities, across 14 economic and social dimensions. Inclusion is a different concept, addressing the extent to which people are included in various aspects of life (or not), and is somewhat more relational, explicitly incorporating issues around belonging and personhood, and including wider social, political, cultural, and attitudinal forms and arenas of exclusion [13,14]. Poverty is often described as an outcome of exclusion but also as a cause, and poor people with disabilities can be included or excluded as much as non-poor people with disabilities. The two concepts are overlapping but not the same [14,15], which is why the coverage in the IDM dataset of some domains and indicators of importance to inclusion is incomplete. Thus, while the IDM dataset includes many aspects that are relevant to the measurement of inclusion, our analysis can also inform us about what data remains missing and how measurement tools could be designed to measure inclusion more effectively and comprehensively at a large scale.

## 2. Materials and Methods

### 2.1. A Framework for Measuring Disability Inclusion

The framework presented in this paper was developed and adapted primarily from the key elements of inclusion as identified in the literature review—8 domains derived from the literature review and a further 3 domains that we can include specifically because of the availability of data from the IDM that is known to be of importance to people with disabilities.

Interpersonal status refers to being seen as a person by others, as worthy of respect, being treated with dignity, and seen as contributing value to the household (this domain is called ‘being accepted’ by Hall and ‘interpersonal status’ by Ikäheimo). Personal relationships are about relationships with family and friends and being able to reciprocate as a sign that what one offers is of value (i.e., contributing to agency and recognition), free from stigma and discrimination (Hall’s ‘having relationships, also overlapping with Ikäheimo’s interpersonal status). Being involved entails having wider social relationships with the broader community, beyond friends and family (Hall’s involvement in activities). Living conditions refers to the basic needs of individuals being met with dignity and without shame (Hall’s adequate living accommodations). Economic opportunity and contributions include a range of economic and other contributions (i.e., paid and unpaid activities) made by the individual to the household and its operation. This framework includes a wider range of contributions than is typically recognized, and which is usually measured as ‘employment’ only (e.g., in Hall). Support systems include formal support received from service providers and is usually provided at a cost, and informal support which is usually unpaid and provided by family, friends, and/or community (Hall’s support systems). Institutional status refers to the formal recognition of the individual by the state/government, which may affect whether states’ obligations towards their citizens can be met (e.g., social assistance/welfare, health, education) (Ikäheimo’s domain of institutional status). For example, in South Africa a social assistance grant is only provided to people with a formal identity document. Voice is about participation in decision making processes affecting the individual (Ikäheimo’s domain of interpersonal status and being seen as worthy). 

Three additional domains are included because they can be populated with data from the IDM South Africa Country Study, and they reflect domains known to be of importance to inclusion generally, as well as to people with disabilities more specifically, underpinning individuals’ ability to be involved in activities and employment (for example). Education entails the formal education received and its quality, while healthcare access refers to both the services accessed (if needed) and their quality. Finally, personal safety refers to individuals’ safety from threats or hazards while undertaking household activities, and their perceptions of their own safety in and around their home.

The individual indicators measured are presented in Section 2.4.

### 2.2. The IDM South Africa Country Study

The IDM recognizes that poverty is experienced at an individual-level, is gender-sensitive, and is multidimensional, and as such, measures it in this way. The 14 dimensions of poverty that the IDM measures were identified largely by participatory work undertaken across 6 countries [16]. Data collection for the IDM South Africa Country Study was undertaken in 2019, and the data from the main national sample, of 8652 individuals, were used in this analysis. For this sample, enumeration areas (EAs) were randomly selected across all 9 provinces, stratified by rural/urban locality. All dwellings within the selected EAs were identified by using satellite imagery to remote sense roofs and were then randomly sampled [17].

Individuals eligible to be interviewed were all of those living in a sampled dwelling who were 16 years and older, able to communicate for themselves, and who were competent to give informed and ongoing consent during the interview. Each individual interview was undertaken with an enumerator of the same gender, in privacy, and in the preferred language of the respondent (choosing from any of the 11 official spoken languages of South Africa). The individual interviews collected data on 14 dimensions of deprivation, on demographic characteristics, and on functioning difficulties, and took, on average, 44 min. Suich et al. provides detailed information about the country study implementation and results [18].

### 2.3. Disability Definition and Measurement

The IDM individual survey tool used the questions in the Washington Group on Disability Statistics Short Set on Functioning (WG SS) of as the measure of disability status [19]. These questions identify people at risk of experiencing disability and its related disadvantages by measuring difficulties people have in various basic activities of seeing, hearing, remembering and concentrating, walking and climbing stairs, communication, and self-care. In this paper, 3 categories of disability severity are determined. The first category is people with no disabilities, who reported ‘no difficulty’ on any functioning or ‘some difficulty’ for only one activity—6952 individuals. The second category is people with ‘mild disabilities’—812 individuals reporting ‘some difficulty’ for between 2 and 4 of the 6 activities but not reporting ‘a lot of difficulty’ or ‘cannot do at all’ for any. The third category is people with moderate or severe disabilities (‘moderate’)—888 individuals reporting ‘a lot of difficulty’ or ‘cannot do at all’ for at least 1 of the 6 activities, or ‘some difficulty’ for 5 or more of the 6 activities. This is a slightly modified version of the recommended cutoff for disability statistics reporting [20]. We have included the category of 5 or more ‘some difficulty’ responses as indicating a moderate to severe disability.

The demographic characteristics of the sample are presented in Table 1 for each of these 3 disability severity categories and the whole sample. These data were not weighted to be fully representative of the South African population, and therefore caution should be exercised in making inferences from these results to the wider South African population.

Few individuals interviewed stated that the functioning difficulties they reported were present at birth (Table 2), in particular for those over 65 years—most of whom appear to be experiencing age-related functioning difficulties.

### 2.4. Selection of Indicators to Populate the Framework

The proposed framework describes the domains of inclusion identified as important for the measurement of inclusion, and Table 3 describes these domains and the indicators selected within each of the domains, as drawn from the IDM South Africa Country Study. The questions used in the IDM individual survey tool were developed specifically to measure deprivation at the individual level and therefore do not overlap exactly with measures of inclusion; thus, some domains can be more comprehensively illustrated using the IDM data than others in the reported results. For example, in the interpersonal status domain, 2 indicators were selected, referring to respect and value associated with unpaid domestic and care work. A specifically designed inclusion measure would include additional aspects of this domain, such as being addressed directly rather than through one’s personal assistant or carer and being listened to when contributing to a discussion, which are not available in the IDM data. Other domains include a range of indicators rarely measured elsewhere. For example, the living conditions domain includes basic clothing and footwear ownership and quality, as well as ownership of bedding materials, which have rarely been measured elsewhere, but can make important contributions to individuals’ dignity. As noted above, we expanded the conceptualization of what is typically identified as the domain of ‘employment’, which typically refers only narrowly to whether an individual is in paid employment. A wider range of economic and other contributions can be determined using the IDM data—including contributions necessary to the running of a household and which may enable other members of the household to undertake paid employment or income-generating activities [21,22], and which are recognized as an important component of household economies. As a result, we renamed this domain ‘economic opportunities and contributions’.

### 2.5. Data Analysis

The results below are presented as the proportion of the sample that falls into each response category. All data in the tables are reported only for those who provided a relevant response to each question, excluding those for whom a question may not be relevant (e.g., those who did not do any unpaid domestic and care work were not asked whether that work was valued by household members). Those few individuals who refused to answer any one question were also excluded; of the 75 survey questions used to create these indicators, 71 had refusal rates of less than 1%. The four questions with refusal rates exceeding 1% were those assessing functional literacy and numeracy, which asked respondents to complete reading and writing tasks and 2 mathematical calculations, for which refusal to answer ranged between 3.8 and 7.1%. Exact *p*-values from chi-squared tests are reported in the Results section tables in order to indicate where statistically significant differences occur between subgroups, using Benjamini–Hochberg adjustments for multiple simultaneous comparisons to control for the false discovery rate (i.e., the number of false-positive results increasing with the number of tests).

## 3. Results

### 3.1. Interpersonal Status

The two indicators in the interpersonal status domain relate solely to respect associated with unpaid domestic and care work (and are only asked of those individuals who reported doing this type of work). As shown in Table 4, there were no statistically significant differences found between the three groups in the levels of respect received.

### 3.2. Personal Relationships

The data for the personal relationships domain, presented in Table 5, show the expected pattern—the proportion of individuals who were able to reciprocate ‘only sometimes’ and ‘never’ was higher for those with mild and moderate disabilities than for individuals with no disabilities.

### 3.3. Being Involved

Table 6 shows the results for the being involved domain. Relatively low levels of all three groups were able to attend community events always or most of the time. The proportion of individuals who reported feeling excluded from these events increased with the severity of disability.

### 3.4. Living Conditions

The living conditions domain had the largest number of indicators of any domain presented in this paper, and the results for each can be seen in Table 7. For all but three of the indicators, the worst outcomes were most frequently realized by those with moderate disabilities, followed by those with mild disabilities and the best outcomes most likely for individuals with no disabilities—across food, water and energy security, basic clothing and footwear, and bedding.

There were three differences from this pattern—higher proportions of those with moderate disabilities than those with mild or no disabilities feared eviction, there were no differences between the three groups with respect to the type of toilet facilities used at home, and a smaller proportion of those with mild disabilities had a modified toilet (compared to those with moderate and no disabilities).

Several issues can be observed from these data. The first is the high levels of food insecurity amongst all three groups—of the overall sample, only 36.2% are food-secure, and severe food insecurity was found to worsen significantly with increasing severity of disability. What is not available in these results is a breakdown of food security for individuals within a household to see if there are intra-household differences which could be explained by disability severity.

Ownership and quality of clothing and footwear are rarely measured, but speak to issues of dignity, respect, and opportunity, and these data illustrate that as disability severity increases, there is a declining proportion of those who own the most basic levels of clothing and footwear, and a decline in the quality of that basic clothing and footwear.

The outcomes for bedding ownership were also found to decline with increasing severity of disability—one in five of those with no disabilities reported not having enough bedding to sleep comfortably (21.1%), compared to 3 in 10 of those with mild disabilities (29.7%) and slightly more for those with moderate disabilities (34.7%).

### 3.5. Economic Opportunity and Contributions

The results for each of the indicators in the economic opportunity and contribution domain, including labor force status and the various indicators demonstrating contributions to the household, can be seen in Table 8. 

In understanding employment status using these data, one must remember that all respondents—regardless of age—were given a categorization for this indicator. That is, even those 65 years and older were included, which is unlike most other statistics dealing with employment status and labor force participation. The category ‘not in the labor force (by choice)’ includes all individuals of legal working age (i.e., younger than 65) who reported being in full time education, individuals of any age who reported they did not need or want to work, and individuals 65 years and older who were retired. The category not in the labor force (exclusion) included individuals who reported not being in the labor force because they were too busy with domestic/caring duties, were unable to work, or were ‘discouraged workers’ (i.e., had given up looking for work because it was too hard to find).

The results show the anticipated pattern—with the lowest rates of exclusion from the labor force amongst those without disabilities and the highest rates amongst those with moderate to severe disabilities. It is not clear why higher proportions of those without disabilities are unemployed, with the proportion declining progressively with disability severity; perhaps those with disabilities are more discouraged (and thus categorized as not in the labor force by exclusion). For the transport indicator, the worst outcomes are experienced by those with mild disabilities, and the best outcomes by those with no disabilities. 

The indicators assessing contributions to the household examine whether individuals do unpaid domestic and care work within the household by whether they spent time the previous day on-call (i.e., responsible for a child under 13, or for caring for a sick, disabled, or elderly person) and whether they collected fuel sources and/or water for the household (for those that relied on sources not delivered to the dwelling). Overall, these indicators show the important contributions made by people with disabilities to the day-to-day running of households.

Though the patterns are different for each indicator, it can be seen that significant contributions to the household are being made by all three disability status groups—with approximately half or more of each group undertaking these four activities. Those with no disabilities were found to have the highest proportion doing unpaid care and domestic work and collecting fuel/energy from outside the dwelling, with slightly lower proportions for those with mild and moderate disabilities. The pattern reversed for on-call time (with the highest proportion being those with moderate disabilities), and there were no differences between the three groups in the collection of water from outside the dwelling.

### 3.6. Support Systems (Formal and Informal Support)

The first two indicators of the support systems domain can be seen in Table 9. The third—which examined individuals’ receipt of the old-age and disability social assistance grants—were found to be further disaggregated by disability status and age group, as seen in Table 10. 

Just over 1 in 10 individuals without disabilities require support but do not receive enough of it (10.8%), compared to 2 in 10 with mild disabilities (19.9%), and one-quarter of those with moderate disabilities (25.6%). Just under half of those with mild and moderate disabilities reported having a carer in the household (40.4% and 43.6%, respectively).

Table 10 highlights the relatively low proportion of individuals reporting functioning difficulties using the WG SS but receiving a disability grant. Note that eligibility rules for social grants mean that all respondents over 60 should receive an old-age pension rather than a disability grant, which suggests a far more effective distribution.

### 3.7. Institutional Status

The indicators of the institutional status domain are current possession of a South African identity document (ID) and/or of a birth certificate, as shown in Table 11. The most accurate interpretation of these indicators is representing ‘current’ institutional personhood, as the questions asked whether individuals possessed an ID document or birth certificate at the time of the survey (i.e., not whether they had ever had one, even if they did not possess one in 2019). The two indicators for this domain move in opposite directions—those with disabilities are more likely to have an ID document, but far less likely to currently be in possession of a birth certificate. This is partly age-related (i.e., older people are less likely to currently have a birth certificate) and likely also because of difficulties with secure storage over the long term.

### 3.8. Voice

The two indicators of the voice domain are shown in Table 12, and the results for both indicators ran in the opposite direction to virtually all other indicators and domains considered. Unexpectedly, individuals with mild and moderate disabilities were found to have better outcomes in both indicators than do individuals with no disabilities. However, there were similar proportions (close to one-quarter) of each group that were excluded from participating in local decision making and from voting freely. This could be a particularity of the South African party political context and the strength of the disability rights movement in South Africa that ensured that disability was specifically included in the South African Constitution [24].

### 3.9. Education

The differences between the three groups in education levels can be seen in Table 1—those with no disabilities were found to be more likely to have higher levels of education, whilst there were very large proportions of those with mild and moderate disabilities who were found to have completed only primary school or less. The remaining indicators of the education domain are functional literacy and functional numeracy, as shown in Table 13.

Far higher proportions of individuals with no disabilities are classified as functionally literate or functionally numerate than those with mild or moderate disabilities. Note that one of the eligibility criteria for participating in the IDM survey was to be able answer questions for themselves, which is likely to bias this sample toward those with less rather than more severe disabilities. Note also that there are important (negative) correlations between functional literacy and numeracy and age, which are related—at least in part—to the lack of education and educational quality received by older citizens.

### 3.10. Healthcare Access

The indicators used in the healthcare access domain are shown in Table 14, which relate to accessing healthcare (excluding prenatal and birthing care) in South Africa in the 12 months prior to the survey (or the reasons why not), and measures of respectful treatment and communication difficulties.

The differences between the three groups were found to be significant for three of the four indicators. Far higher proportions of those with mild and moderate disabilities sought healthcare than those without, and of those who sought healthcare, those with moderate disabilities were most likely to not receive respectful treatment, although the numbers were small. There were no significant differences found between the groups with respect to problems with communication with healthcare professionals, though a small minority did experience this problem.

Of those who did not access healthcare, individuals with mild and moderate disabilities were more likely to have reported feeling excluded from accessing healthcare than those with no disabilities—whether because healthcare was too costly, was too far away, there was no transport to get there, the respondent was too embarrassed to seek healthcare, the provider refused to treat the individual, or they felt vulnerable to discrimination. The number of excluded respondents were too small to analyze separately.

### 3.11. Personal Safety

The four indicators in the personal safety domain are shown in Table 15. For three of the four indicators in this domain, outcomes deteriorated with increasing severity of disability, as for most other domains, with those with moderate disabilities experiencing overall the lowest levels of personal safety (there was no difference between the three groups for fuel collection threats). Note that the figures reported for those facing hazards while collecting water and fuel from outside the dwelling are reported only for those individuals who reported being responsible for this activity.

## 4. Discussion

### 4.1. Associations between Disability Severity and Inclusion

An objective of this paper was to conduct an initial exploratory analysis to operationalize the proposed framework to assess disability inclusion and to examine associations between important domains of inclusion and disability severity. Across the 11 domains described, we identified 40 indicators measuring aspects of inclusion. Of these, only six showed no statistically significant differences between those with no disabilities, mild disabilities, and moderate disabilities. These included the two indicators in the interpersonal status domain (asking about being humiliated while doing unpaid domestic and care work and such work being valued), water collection responsibility, communication problems with healthcare worker, fuel collection hazards (if required to collect fuel), and having a toilet facility at home.

Of the remaining 34 indicators, by and large those with no disabilities had the best outcomes, followed by those with mild disabilities, and the worst outcomes were reported by those with moderate and severe disabilities. In a few cases, there were no differences between those with mild and moderate disabilities, but those with no disabilities had better outcomes. This analysis therefore indicates a negative relationship between disability severity and inclusion—those with disabilities experienced a comparative lack of inclusion (worse outcomes) compared to those without disabilities—even though, in some cases, those with no disabilities also had poor outcomes.

However, there were some unusual or unexpected patterns, where the outcomes were better for those with disabilities than those with no disabilities. One example is in the voice domain, measured using two indicators—the ability to vote freely and without coercion, and participation in local decision making. While there was generally a low level of participation in local decision making by all groups, it was lowest for people with no disabilities—people with disabilities were found to be slightly more likely to participate in such processes. The strong presence of disabled people’s organizations in South Africa may contribute to people with disabilities feeling part of the activities and running of these organizations [24]. The higher levels of voting, and voting freely, may be linked to the activities of the South African Independent Electoral Commission to increase voting participation by people with disabilities. While these suggest positive trends in inclusion, further research is needed to better understand the meaning and drivers of these results.

There are some results that are of particular interest, such as those for food insecurity and for access to clothing and footwear. These are not commonly reported outcomes in relation to disability at the population level and highlight an important area of disadvantage. The results show very high levels of food insecurity for all three groups, but outcomes were found to be worse for those with mild disabilities and worst for those with moderate or severe disabilities. More than half of those with mild disabilities, and almost two-thirds of those with moderate disabilities experienced severe food insecurity, with these data being collected before COVID-19 lockdowns and associated economic impacts. Studies in the USA, Canada, and South Korea show that people with disabilities are more likely to be or live in households that are food-insecure [25,26,27]. While there seems to be little literature reporting on this in low- and middle-income countries, there is some emerging evidence of high food insecurity among people with severe mental illness in Ethiopia [28].

A very rarely measured indicator in relation to disability (or indeed with respect to poverty more generally) is that of access to adequate clothing and footwear. The IDM measured whether individuals owned at least two complete changes of clothing and footwear or not. There are high proportions from all three groups experiencing limited ownership of clothing and footwear, including one in five of those with mild disabilities and one in four of those with moderate or severe disabilities. The standard applied is low, and a lack of even this level of clothing and footwear has important implications for people’s ability to move around in public with dignity and without shame, as well as for employment possibilities, among other things. A scan of the published literature confirms the lack of reporting of access to clothing and footwear as a factor in experiences of people with disabilities. A scoping review on the role of clothing on participation of people with physical disabilities shows that clothing design is also an important determinant of participation [29]. Clothing that is not designed to accommodate a physical disability will limit participation. However, other factors, such as poverty or neglect, may limit the number of changes of clothing a person with disabilities owns.

In the literature, one of the main indicators of inclusion is paid employment, but the IDM dataset provides us with an opportunity to look beyond employment and to consider additional contributions which are often ignored in conventional statistics because these activities are typically unpaid. As anticipated, employment levels are lower for those with disabilities than those without, though all three groups have poor outcomes of low employment levels. However, significant contributions are made by all groups in terms of unpaid activities—for example, in unpaid domestic and care work, and in fuel and water collection (from outside the dwelling). While there are often higher proportions of people without disabilities that do these activities, this is not true for every activity, and there are substantial proportions of people with mild and moderate disabilities making these types of contributions to their households. If being seen as ‘productive’ is a marker of inclusion, we can start to describe productivity more broadly than employment and increase the visibility of these contributions that are so often ignored. This would be in line with global trends of including unpaid work (usually performed by women) as part of the global economy [21,22].

This initial analysis does not consider a range of other factors that could provide other explanations for these significant differences—for example, the role of household poverty in accessing food and adequate clothing for everyone in the household and not just the person with disabilities in that household; the role of age in providing unpaid care where older people, who are more likely to have disabilities, are required to provide care for their grandchildren—a common occurrence in South African households [30]. The IDM data are individual level outcomes and the analysis conducted to date does not yet provide a comparison of levels of intra-household differences between inclusion of a person with disabilities in relation to non-disabled members of their own household. Such analyses could improve our understanding of the potential effect of these variables on the measures of inclusion.

The cut-offs chosen for each indicator, determining whether an individual is categorized as included or not, were described for each variable in a way that seemed to differentiate between a strongly positive (inclusion) and a strongly negative outcome (exclusion) as set out for each indicator. However, these cut-offs have not been tested and would benefit from further scrutiny.

As noted above, the IDM data used are unweighted and only those people who were able to answer for themselves and give informed consent were recruited into the IDM survey (i.e., excluding people with more severe disabilities who were not able to respond for themselves); thus, caution should be exercised in making inferences from these results to the wider South African population with disabilities. Furthermore, given the high levels of poverty and extremely high levels of inequality in South Africa, which are atypical features of middle-income countries, these results are not thought to be widely generalizable to other middle-income countries.

### 4.2. Limitations of Using Existing Datasets for Measuring Inclusion

One of the objectives of this analysis was to assess the feasibility of using an existing dataset for an allied but different purpose—the domains of inclusion identified in the framework proposed for measuring disability inclusion were populated with IDM data, where the data measured aspects of these domains. Given the high costs of collecting data on sufficient numbers of respondents with disabilities, it makes sense to economize on data collection activities where possible.

This study highlights a number of limitations associated with using existing datasets for a different purpose than they were originally designed, as described using examples from our use of IDM data, designed to measure individual-level multidimensional deprivation and repurposed to populate the framework for measuring disability inclusion.

The first is that a number of the domains of inclusion were not comprehensively measured because the IDM survey tools do not include questions covering the measurement of all aspects of these domains. Examples include the limited data that can be used to populate the domains of interpersonal status, personal relationships, and being involved, and which therefore make it difficult to come to meaningful conclusions about the relationships between disability severity and inclusion in these domains.

The second limitation occurs when some data have to be excluded because they are insufficient or incomplete. For example, the IDM dataset contains information about whether individuals own their dwelling and/or the land on which it sits, but this could only be used as a measure of inclusion if it was supplemented with information on whether individuals choose to own their dwelling, or are somehow prevented from doing so (e.g., renting may be a valid choice, and thus not indicative of exclusion). IDM data are also available regarding the use of assistive devices for those with functioning difficulties. However, these data would need to be supplemented with information about whether individuals had an unmet need for an assistive device to determine whether not using an assistive device was an indicator of a lack of inclusion.

Finally, some of the IDM indicators could potentially be allocated to more than one domain. For example, while the interpersonal status domain should incorporate issues of respect, the healthcare domain also includes an indicator assessing whether individuals were treated respectfully by their healthcare provider.

Each of these limitations would be addressed by specifically designing measures of inclusion—and the necessary survey tools—in order to ensure comprehensive data capture with appropriate coverage of indicators within each domain measured.

### 4.3. Future Directions—Considerations for Monitoring Inclusion for People with Disabilities

It is difficult to use existing data that have been collected for different purpose, but the IDM data do provide evidence of some important trends. The first is that it highlights individual-level information across a range of indicators that are rarely (if ever) available, such as access to adequate clothing and footwear, as well as contributions to the household. Secondly, it shows broad trends (as expected) in the associations between lack of inclusion and severity of disability. Given that the IDM survey tools were designed to be broadly standard across contexts, this analysis could potentially be replicated using other IDM datasets, and the feasibility of using other non-IDM datasets containing relevant information (including an appropriate measure of disability status) using this framework for measuring inclusion could also be examined.

However, survey tools designed specifically to measure inclusion should include not only more detailed questions on many of the domains such as being respected and having dignity, but also measures of the person’s own sense of their level of inclusion. Thus, while this analysis has also demonstrated that it feasible to operationalize this framework for measuring disability inclusion utilizing an existing dataset, attempts to do so would be better served by specifically designed tools.

Analysis of the current domains disaggregated by factors such as age, age of onset, type of disability, and gender is possible, though not undertaken in this paper, and would be recommended to yield a more nuanced understanding of these results on the basis of the IDM data and the factors driving levels of inclusion. Importantly, any specifically designed inclusion measure should ensure that intersectional analyses such as these (and others identified as important) are possible.

Two additional lessons that are relevant to the design of tools to measure inclusion specifically can be drawn from the design process of the IDM. One is the importance of designing a gender-sensitive measure—for example, the IDM collects data on various aspects of menstruation, including the non-attendance of activities because of a lack of sanitary products and/or shame or stigma associated with menstruation, which are collected only for women. A gender-sensitive measure of inclusion should also include data about those issues in each domain that affect women and men differentially.

A second lesson can be drawn from the process of selecting the dimensions of deprivation that are measured by the IDM. The 14 dimensions measured in the IDM were selected, in large part, on the basis of participatory work conducted with poor men and women to identify the dimensions of poverty that they themselves prioritized. To maximize relevance for people with disabilities and their supporting organizations, we recommend that any future work to design specific tools to measure disability inclusion should also include participatory work with people with disabilities designed so they identify and prioritize the key domains and indicators of inclusion that are important to measure. Such a process would be likely to highlight important indicators (and perhaps domains) of inclusion that have not been considered in this analysis.

## 5. Conclusions

The objectives of this paper were threefold: the first was to propose a framework of the key domains of inclusion that should be included in large-scale studies measuring levels of inclusion of people with disabilities. The second was to operationalize the framework using an existing dataset, from the IDM South Africa Country Study, and thirdly, to assess the feasibility of using existing datasets to measure inclusion. The purpose of operationalizing the framework is to improve our understanding of the associations between disability severity and levels of inclusion.

Do the IDM data allow us to measure inclusion on the basis of the proposed framework? This analysis suggests that there are important and interesting elements that show promise and that could be developed further to create better measures of inclusion. Overall, the data show the expected (negative) relationship between social inclusion outcomes and disability severity. However, because the IDM was designed to measure deprivation, it is not as comprehensive with respect to indicators of social inclusion as measures designed specifically to understand inclusion would be. That means we are missing data on some important issues, for example, about the recognition of people with disabilities by others, as well as on issues around individuals’ social life.

This analysis shows that the use of data collected for one purpose can have utility for alternative purposes, though there are limitations of the approach. These limitations would be overcome by using specifically designed tools, but in the absence of such specifically designed tools and given the high costs of data collection, it makes sense to examine the feasibility of using existing datasets containing relevant data for alternative purposes.

## Figures and Tables

**Table 1 ijerph-18-04431-t001:** Demographic characteristics in terms of disability status from the main study (%).

	None	Mild	Moderate	Overall
Total cases	6952	812	888	8652
Gender				
Male	46.6	34.6	24.3	43.2
Female	53.4	65.4	75.7	56.8
Age				
16–24	28.2	9.7	6.9	24.3
25–64	66.1	57.3	59.1	64.6
65+	5.7	33	34	11.2
Population group				
Black African	84.0	83.9	83.3	83.9
Coloured	12.7	12.9	14.6	12.9
Indian or Asian	1.1	1.1	0.9	1.0
White	2.2	2.1	1.1	2.1
Educational completion				
Matriculation or higher	45.7	23.6	20.4	41.0
Some secondary schooling	39.0	33.9	30.6	37.7
Primary or less	15.3	42.5	49.0	21.3

**Table 2 ijerph-18-04431-t002:** Age of onset of disability for those with mild and moderate disabilities in terms of gender and age (%).

	Male			Female		
	16–24	25–64	65+	16–24	25–64	65+
At birth	11.3	4.3	0	3.4	2.8	1.0
Childhood/school age	64.2	13.2	1.2	63.2	9.2	0.7
Early adulthood (±18–29)	22.6	17.4	2.5	32.2	14.7	3.2
Later (30+)	1.9	65.1	96.3	1.1	73.2	95.1
Total cases	53	281	161	87	706	409

**Table 3 ijerph-18-04431-t003:** Domains of inclusion and indicators used to measure aspects of those domains (from the Individual Deprivation Measure (IDM) dataset).

Domain of Inclusion	Indicator (Drawn from the IDM South Africa Country Study)	Concept Measured
Interpersonal status	Unpaid domestic and care work humiliation	Whether the respondent was subject to humiliating treatment while doing unpaid domestic and care work
	Unpaid domestic and care work value	Whether household members of the respondent value the unpaid domestic and care work they do
Personal relationships	Ability to reciprocate support	Frequency and ability to reciprocate support received
Being involved	Community event inclusion (#)	Frequency and ability to participate in community events (e.g., religious activities, ceremonies, or festivals)
Living conditions	Food security (#) *	Degree of food (in)security in not having sufficient and nutritious food
	Drinking water source and reliability (#)	Type (improved/unimproved) and reliability of drinking water source
	Domestic water source and reliability (#)	Type (improved/unimproved) and reliability of domestic water source
	Cooking energy source and reliability (#)	Type (clean/polluting) and reliability of cooking energy source
	Lighting energy source and reliability (#)	Type (clean/polluting) and reliability of heating energy source
	Heating energy source and reliability (#)	Type (clean/polluting) and reliability of heating energy source
	Home toilet facilities (#)	Type of toilet facility (improved/unimproved)
	Toilet modifications	If toilet facility is (partially) modified to accommodate physical needs
	Basic clothing and footwear ownership (#) *	Ownership of two complete sets of basic clothing and footwear
	Basic acceptability and protection (#) *	Acceptability and protection of basic clothing and footwear
	Bedding ownership	Ownership of sufficient bedding materials
	Eviction concern	Fear of eviction from accommodation
Economic opportunity and contributions	Labor force status (#)	Labor force status (all respondents)
	Unpaid domestic and care work	Unpaid domestic and care work undertaken at home
	Fuel collection responsibility	Responsibility for collecting fuel sources from outside the home (if necessary)
	Water collection responsibility	Responsibility for collecting water sources from outside the home (if necessary)
	On call time (#)	Responsibility for caring for a child under 13 and/or a sick, elderly, or disabled person (the previous day)
	Public transport availability and affordability (#)	Availability and affordability of public transport
Support systems (formal and informal support)	Support availability	Need for help to meet basic needs and support and frequency of receiving it
	Old-age pension	Receipt of an old-age pension (if relevant)
	Disability grant	Receipt of a disability grant (if relevant)
Institutional status	Identity document	Current possession of a South African identity document
	Birth certificate	Current possession of a birth certificate
Voice	Local decision-making inclusion (#)	Frequency and ability to participate in local decision making
	Voting inclusion	Freedom to vote and vote freely
Education	Educational completion	Level of education achieved
	Basic literacy (#)	Functional literacy (ability to read and to write in an official language)
	Basic numeracy (#)	Functional numeracy (ability to complete two simple mathematical calculations)
Healthcare access	Healthcare access (#)	Accessed healthcare in South Africa (or reason why not)
	Healthcare communication	Communication difficulties associated with health treatment
	Respectful treatment	Respectful treatment from healthcare workers
Personal safety	Fuel collection hazards	Experienced hazards/threats while collecting fuel outside the home (for those responsible)
	Water collection hazards	Experienced hazards/threats while collecting water outside the home (for those responsible)
	Safety in the neighborhood	Perceived safety of walking alone in the neighborhood after dark
	Safety at home	Perceived safety of being at home alone after dark

(#) indicator constructed from more than one survey question. * an indicator constructed using the IDM scoring methodology [23].

**Table 4 ijerph-18-04431-t004:** Interpersonal status domain indicators in terms of disability status.

Variable	Level	None	Mild	Moderate	*p*-Value	Overall
Humiliating treatment whilst carrying out unpaid domestic and care work	No	97.2	96.2	95.9	0.08942	97.0
	Yes	2.8	3.8	4.1		3.0
		6022	655	708		7385
Unpaid domestic and care work valued by household members	Yes	87.0	87.6	88.7	0.41056	87.2
	No	13.0	12.4	11.3		12.8
		6039	654	708		7401

**Table 5 ijerph-18-04431-t005:** Personal relationships domain indicator in terms of disability status.

Variable		None	Mild	Moderate	*p*-Value	Overall
Reciprocation	Can reciprocate always or most of the time	43.5	39.9	37.0	0.00031	42.5
	Can reciprocate only some of the time or never	56.5	60.1	63.0		57.5
		6918	809	881		8608

**Table 6 ijerph-18-04431-t006:** Participation in community events domain indicator in terms of disability status.

Variable	Level	None	Mild	Moderate	*p*-Value	Overall
Participation in community events	Always or sometimes attended community events OR there was no event to attend	56.7	58.3	46.6	0.00001	55.9
	Rarely attended OR did not attend because individual was too busy, too sick, or not interested in doing so	32.6	27.8	34.4		32.3
	Rarely or never attended because they were prevented from doing so or excluded	10.7	13.9	19.0		11.8
		6885	803	879		8567

**Table 7 ijerph-18-04431-t007:** Living conditions domain indicators in terms of disability status.

Variable	Level	None	Mild	Moderate	*p*-Value	Overall
Food security	Food secure	39.1	26.6	22.2	0.00001	36.2
	Mild or moderate food insecurity	20.9	18.3	14.5		20.0
	Severe food insecurity	39.9	55.0	63.3		43.7
		6952	812	888		8652
Drinking water	Improved drinking water and enough to meet needs always or most of the time	80.8	72.0	69.7	0.00001	78.8
	Improved drinking water and enough to meet needs some of the time or never OR unimproved drinking water and enough to meet needs always or most of the time	17.2	24.8	27.3		19.0
	Unimproved drinking water source and enough to meet needs some of the time or never	2.0	3.2	3.0		2.2
		6950	812	888		8650
Domestic water	Improved domestic water and enough to meet needs always or most of the time	82.2	72.0	68.4	0.00001	79.8
	Improved domestic water and enough to meet needs some of the time or never OR unimproved domestic water and enough to meet needs always or most of the time	17.6	27.8	31.5		20.0
	Unimproved domestic water source and enough to meet needs some of the time or never	0.2	0.1	0.1		0.2
		6948	812	887		8647
Cooking energy	Clean cooking energy and enough to meet needs always or most of the time	68.6	59.2	50.1	0.00001	65.8
	Clean cooking energy and enough to meet needs some of the time or never OR unclean cooking energy and enough to meet needs always or most of the time	24.4	31.2	42.5		26.9
	Unclean cooking energy and enough to meet needs some of the time or never	7.0	9.6	7.4		7.3
		6948	812	887		8647
Lighting energy	Clean lighting energy and enough to meet needs always or most of the time	78.9	66.4	58.0	0.00001	75.6
	Clean lighting energy and enough to meet needs some of the time or never OR unclean lighting energy and enough to meet needs always or most of the time	18.2	29.6	40.1		21.5
	Unclean lighting energy and enough to meet needs some of the time or never	3.0	4.1	1.9		3.0
		6947	812	888		8647
Heating energy	Clean heating energy and enough to meet needs always or most of the time	57.7	54.3	43.0	0.00001	55.9
	Clean heating energy and enough to meet needs some of the time or never OR unclean heating energy and enough to meet needs always or most of the time	23.3	25.7	33.9		24.6
	Unclean heating energy and enough to meet needs some of the time or never	19.0	20.0	23.1		19.5
		6924	806	883		8613
Toilet facility (at home)	Improved toilet facility	85.9	82.9	83.6	0.0166	85.3
	Unimproved toilet facility	12.3	15.8	14.8		12.9
	No toilet facility	1.9	1.4	1.6		1.8
		6951	811	885		8647
Modified toilet facility	Modified to accommodate physical needs	73.8	60.6	71.4	0.00001	69.0
	Partly modified to accommodate physical needs	5.9	11.2	6.5		7.7
	Not modified	20.4	28.1	22.1		23.3
		904	747	802		2453
Ownership of basic clothing and footwear	Own two changes of clothes and two pairs of footwear	86.8	79.2	73.5	0.00001	84.7
	Do not own two changes of clothes and two pairs of footwear	13.2	20.8	26.5		15.3
		6952	812	888		8652
Basic clothing and footwear quality	Basic clothing is acceptable always OR most of the time AND protection is excellent OR good	71.7	58.5	52.0	0.00001	68.5
	Basic clothing ownership is always OR most of the time AND provides some or no protection; basic clothing is acceptable some of the time OR never AND protection is excellent OR good	17.9	24.6	25.8		19.3
	Basic clothing is acceptable some of the time OR never AND provides some or no protection	10.4	16.9	22.2		12.2
		6952	812	888		8652
Ownership of sufficient bedding	Own sufficient bedding to sleep comfortably	78.9	70.3	65.3	0.00001	76.7
	Do not own sufficient bedding to sleep comfortably	21.1	29.7	34.7		23.3
		6943	811	887		8641
Fear of eviction	Did not fear eviction	90.5	90.1	87.3	0.00926	90.2
	Feared eviction	9.5	9.9	12.7		9.8
		6944	808	888		8640

**Table 8 ijerph-18-04431-t008:** Economic opportunity and contributions domain indicators in terms of disability status.

Variable	Level	None	Mild	Moderate	*p*-Value	Overall
Labor force status	Employed	39.2	24.7	20.8	0.00001	35.9
	Unemployed	27.0	15.8	13.8		24.6
	Not in the labor force (choice)	24.0	45.7	40.7		27.8
	Not in the labor force (exclusion)	9.8	13.7	24.7		11.7
		6915	809	886		8610
Transport	Public/mass transport available and affordable always or most of the time	43.5	32.6	38.3	0.00001	41.9
	Public/mass transport available and affordable some of the time OR some public/mass transport which is affordable always or most of the time	35.2	37.0	33.0		35.2
	Some public/mass transport which is affordable some of the time or never OR no public/mass transport available	21.3	30.5	28.8		22.9
		6945	811	886		8642
Unpaid domestic and care work	Performed unpaid domestic and care work	87.2	80.9	80.3	0.00001	85.9
	No unpaid domestic and care work	12.8	19.1	19.7		14.1
		6942	812	884		8638
On-call	Spent time responsible for a child under 13 and/or a sick, elderly, or disabled person	46.6	51.5	50.8	0.00404	47.5
	Did not spend time responsible for a child under 13 and/or a sick, elderly, or disabled person	53.4	48.5	49.2		52.5
		6952	812	888		8652
Fuel collection responsibility	Collected fuel from outside the dwelling	69.7	68.2	59.7	0.00001	68.5
	Did not collect fuel from outside the dwelling	30.3	31.8	40.3		31.5
		6939	812	885		8636
Water collection responsibility	Collected water from outside the dwelling	58.5	62.1	60.0	0.12471	59.0
	Did not collect water from outside the dwelling	41.5	37.9	40.0		41.0
		6947	810	887		8644

**Table 9 ijerph-18-04431-t009:** Support systems domain indicators in terms of disability status.

Variable	Level	None	Mild	Moderate	*p*-Value	Overall
Support	Do not need support OR do need support, but get enough always or most of the time	89.2	80.1	74.4	0.00001	86.9
	Need support, and have enough only some of the time or never	10.8	19.9	25.6		13.1
		6931	809	884		8624
Carer in household	Yes	32.6	40.4	43.6	0.00001	37.8
	No	67.4	59.6	56.4		62.2
		1360	812	888		3060

**Table 10 ijerph-18-04431-t010:** Social assistance domain indicator in terms of disability status and age group.

	None			Mild			Moderate		
	16–24	25–64	65+	16–24	25–64	65+	16–24	25–64	65+
Old-age grant									
No	100	96.2	9.9	100	81.5	8.6	100	84.1	5.6
Yes	-	3.8	90.1	-	18.5	91.4	-	14.9	94.4
Total cases	1961	4596	395	79	465	268	61	525	302
Disability grant									
No	99.6	98.8	99.5	98.7	94.2	98.5	93.4	85.1	99.0
Yes	0.4	1.2	0.5 *	1.3	5.8	1.5 *	6.6	13.9 *	1.0
Total cases	1961	4596	395	79	465	268	61	525	302

* It is likely that these individuals (who are not eligible for a disability grant) misreported the specific type of social assistance they received. On turning 60 years of age, recipients of the disability grant are automatically transferred to an old-age grant.

**Table 11 ijerph-18-04431-t011:** Institutional personhood domain indicators in terms of disability status.

Variable	Level	None	Mild	Moderate	*p*-Value	Overall
Identity document	Current possession of a South African identity document	89.0	95.7	97.0	0.00001	90.4
	No current possession of a South African identity document	11.0	4.3	3.0		9.6
		6950	811	887		8648
Birth certificate	Current possession of a birth certificate	66.2	48.6	44.8	0.00001	62.4
	No current possession of a birth certificate	33.8	51.4	55.2		37.6
		6761	774	849		8384

**Table 12 ijerph-18-04431-t012:** Voice domain indicators in terms of disability status.

Variable	Level	None	Mild	Moderate	*p*-Value	Overall
Participating in local decision making	Participated in local decision making	26.6	36.7	33.6	0.00001	28.3
	Did not participate because too busy; not interested; no process to participate in	46.3	37.3	41.1		45.0
	Did not participate because excluded (not invited; afraid/uncomfortable; do not trust the leaders; not appropriate for me)	27.1	26.0	25.3		26.8
		6872	799	881		8552
Voting freely	Voted and free to choose who to vote for	53.3	66.7	69.6	0.00001	56.2
	Did not vote, because not interested, not old enough to vote, OR not a citizen	17.4	9.2	6.7		15.5
	Voted but not free to choose who to vote for OR did not vote, for all other reasons	29.3	24.2	23.8		28.3
		6915	807	887		8609

**Table 13 ijerph-18-04431-t013:** Education domain indicators in terms of disability status.

Variable	Level	None	Mild	Moderate	*p*-Value	Overall
Functional literacy	Able to read and write to a basic level in an official language	77.9	50.9	48.3	0.00001	72.3
	Able to read or write to a basic level in an official language	16.1	22.7	24.5		17.6
	Not able to read or write to a basic level in an official language	6.0	26.4	27.2		10.1
		6542	770	845		8157
Functional numeracy	Able to correctly answer two mathematics problems	67.0	41.8	41.8	0.00001	62.2
	Able to answer one mathematical problem	20.8	22.8	23.2		21.3
	Unable to answer mathematical problems	12.1	35.4	35.0		16.5
		6506	732	801		8039

**Table 14 ijerph-18-04431-t014:** Healthcare domain indicators in terms of disability status.

Variable	Level	None	Mild	Moderate	*p*-Value	Overall
Sought healthcare in previous 12 months (in RSA)	No	46.9	26.0	27.7	0.00001	43.0
	Yes	53.1	74.0	72.3		57.0
		6944	812	887		8643
Reason for not seeking healthcare	Did not need or want	98.4	91.0	93.9	0.00001	97.7
	Excluded	1.6	9.0	6.1		2.3
		3249	211	245		3705
Respectful treatment	Received respectful treatment	92.6	92.7	88.7	0.00352	92.1
	Did not receive respectful treatment	7.4	7.3	11.3		7.9
		3686	600	639		4925
Communication problems	No communication difficulties with healthcare provider	93.8	91.7	91.4	0.0212	93.2
	Communication difficulties with healthcare provider	6.2	8.3	8.6		6.8
		3686	601	639		4926

**Table 15 ijerph-18-04431-t015:** Personal safety domain indicators in terms of disability status.

Variable	Level	None	Mild	Moderate	*p*-Value	Overall
Fuel collection hazards	No hazards while collecting fuel outside the dwelling	90.2	89.9	89.4	0.89499	90.0
	Faced hazards while collecting fuel outside the dwelling	9.8	10.1	10.6		10.0
		2092	257	357		2706
Water collection hazards	No hazards while collecting water outside the dwelling	94.9	95.8	91.2	0.01	94.6
	Faced hazards while collecting water outside the dwelling	5.1	4.2	8.8		5.4
		2852	306	352		3510
Walking alone in the neighborhood after dark	Very safe	6.0	3.6	3.6	0.00001	5.5
	Safe	30.2	25.4	18.4		28.6
	Unsafe	40.3	41.6	33.1		39.7
	Very unsafe	23.5	29.3	44.9		26.2
		6920	798	873		8591
At home by yourself after dark	Very safe	20.6	12.2	16.4	0.00001	19.4
	Safe	54.9	51.7	46.6		53.7
	Unsafe	17.6	26.1	22.9		18.9
	Very unsafe	7.0	10.0	14.1		8.0
		6945	811	884		8640

## Data Availability

The data presented in this study have been lodged with the Australian Data Archive.

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
