# Peer review of "Measuring Disability Inclusion: Feasibility of Using Existing Multidimensional Poverty Data in South Africa"

_ijerph, 2021, doi:10.3390/ijerph18094431_

Round 1

Reviewer 1 Report

I converted the original file to a Word document to give more specific feedback. However, this also means that the format may be significantly different.

Some additional feedback:

There needs to be a more robust conversation around your specific methods (i.e. how did you conduct interviews?, etc.).

In addition, there was a significant amount of detail missing related to the specific data analyses methods you chose to analyze your data.

Author Response

We thank all three reviewers for their comments, and we believe the revisions we have made to the paper in light of these comments have improved its clarity and focus.

We have included a revised version with tracked changes of insertions to facilitate the review of the revised version.

Reviewer 1: Comments and Suggestions for Authors

I converted the original file to a Word document to give more specific feedback. However, this also means that the format may be significantly different.

We did query whether we could have a copy of the Word document described in this comment, believing that it may contain more specific feedback (email to Eloise on 03/03), but with no further information we were not able to address this comment further.

Some additional feedback:

There needs to be a more robust conversation around your specific methods (i.e. how did you conduct interviews?, etc.).

To address this comment, we have clarified and extended Section 2 (Materials and Methods). In particular, we have revised Section 2.2 which describes the IDM South Africa Country Study to describe the sample size, the selection of the sample and eligible individuals. Additional detail has also been provided about the context, content and length of interviews, and question response rates.

In addition, there was a significant amount of detail missing related to the specific data analyses methods you chose to analyze your data.

We have also provided additional information has also been included in Section 2.4 which describes the method used for the analysis of the relationship between disability severity and inclusion. Specifically, Information regarding the data analysis has been added in the last para before Section 3: “The results below are presented as the proportion of the sample that falls into each response category. All data in the tables are reported only for those who provided a relevant response to each question, excluding those for whom a question may not be relevant (e.g. those who did not do any unpaid domestic and care work were not asked whether that work was valued by household members). Those few individuals who refused to answer any one question were also excluded; of the 75 survey questions used to create these indicators, 71 had refusal rates of less than 1%. The four questions with refusal rates exceeding 1% were those assessing functional literacy and numeracy, which asked respondents to complete reading and writing tasks and two mathematical calculations, for which refusal to answer ranged between 3.8–7.1%. Exact p-values from Chi-square tests are reported in the results tables, to indicate where statistically significant differences occur between subgroups, using Benjamini-Hochberg adjustments for multiple simultaneous comparisons to control for the false discovery rate (i.e. the number of false positive results increasing with the number of tests).”

Reviewer 2 Report

The survey is significant as a confirmation of the need for additional state support for individuals with disabilities in South Africa, but the scientific soundness is not clear.

The theoretical review is absent in the article. 

The aim of this paper is defined as 'to explore the feasibility of using IDM South Africa data on multidimensional poverty to monitor levels of inclusion of people with disabilities', but the results show only the comparative analysis of indicators of the quality of life of individuals with and without disabilities without assessing the validity of the method.

In the conclusion there is no inference about a ways forward for further development of IDM, that means that the third objective of the article is not reached.

There are signs of careless performing: some links are not formatted (' Error! Reference source not found' ), at the end of the conclusion there is a phrase 'This section is mandatory. Please summarize the main achievements and/or results in this section.'

Author Response

We thank all three reviewers for their comments, and we believe the revisions we have made to the paper in light of these comments have improved its clarity and focus.

We have included a revised version with tracked changes of insertions to facilitate the review of the revised version.

Reviewer 2:

The survey is significant as a confirmation of the need for additional state support for individuals with disabilities in South Africa, but the scientific soundness is not clear.

In response to this comment (and those of reviewers 1 and 3), we have made revisions to Section 2 (Materials and Methods), in order to improve the clarity of information around the implementation of the IDM study in South Africa (Section 2.2), as well as providing more detailed information about the data analysis methods utilised in the operationalisation of the proposed framework (Section 2.5).

See also the discussion below of the impacts of the revisions we have made to the specific aims and objectives of the paper which improves the justification of the soundness of our approach.  

The theoretical review is absent in the article. 

We have added text below Table 3 that sets out in more detail the conceptual framework that we use to describe our understanding of inclusion in relation to disability as per our review of relevant literature. It’s not clear what theoretical aspects would be most appropriate to further add beyond setting out a framework for inclusion to guide our selection of relevant domains from the IDM South Africa to measure inclusion.   

The aim of this paper is defined as 'to explore the feasibility of using IDM South Africa data on multidimensional poverty to monitor levels of inclusion of people with disabilities', but the results show only the comparative analysis of indicators of the quality of life of individuals with and without disabilities without assessing the validity of the method.

As mentioned above, we have revised the specific aims and objectives of the paper to better reflect the purpose and focus of the paper, and improve the justification of the relevance and soundness of our approach. The revised objectives of the paper can be summarised as proposing a framework for measuring disability inclusion, providing an example of the operationalisation of the framework by using the IDM South Africa data (assessing the feasibility of using existing datasets to populate the framework), thus enabling us to draw conclusions with respect to lessons for the design, effectiveness and comprehensiveness of large-scale quantitative measures of inclusion.

This is strengthened by the additional information provided in Section 2 (Materials and Methods), specifically about details of the implementation of the IDM South Africa Country Study (Section 2.1), and the information provided about our data analysis methods (Section 2.5). 

In the conclusion there is no inference about a ways forward for further development of IDM, that means that the third objective of the article is not reached.

In light of the revision and refinement of the objectives of the paper, we have also revised the text in Sections 4 and 5 (the discussion and conclusions, respectively) to draw stronger links between the revised aims and objectives—in particular the results of operationalising the framework and lessons for large-scale measurement of inclusion, that also addresses this point. 

There are few lessons that are of specific relevance to the further development of the IDM, as that initiative has a specific purpose to develop a measure of gender-sensitive multidimensional deprivation. It does not intend to measure inclusion, though we have used some of the IDM South Africa Country Study data to populate the conceptual framework proposed in the paper to test the notion of using existing data sets rather than having to collect new ones for each specific purpose. However, Section 4.2 has been revised to emphasise the lessons drawn from this study about using an existing dataset for a purpose for which it was not initially designed, and Section 4.3 discusses some of the issues that would need to be considered in the design of tools and measures specifically designed to measure disability inclusion.

There are signs of careless performing: some links are not formatted (' Error! Reference source not found' ), at the end of the conclusion there is a phrase 'This section is mandatory. Please summarize the main achievements and/or results in this section.'

We reviewed our Word version submitted originally and cannot find any reference to bookmark errors. We assume this was an artefact that arose when the document was transformed to a pdf format during the submission.

The phrase at the end of the conclusion was left in error and was from the journal template provided. We have now deleted this as the conclusion had been included.  

Reviewer 3 Report

The article presents findings from a study which used data from the Individual Deprivation Measure for South Africa to measure inclusion of people with mild and moderate/severe disability.  Overall the findings indicated a number of variables in the dataset which could be used for this purpose, but also identified some gaps and limitations.  The article is well written and the methods and findings are clearly presented, and limitations are also extensively discussed.  I only have 2 suggestions for improvement.  Firstly a bit more information about the study should be provided, in particular overall numbers of respondents and numbers of people with mild and moderate disability who responded to the survey, and ideally the response rate as well.  Secondly some more discussion on the relationship between inclusion/exclusion and deprivation would help contextualise these findings.  Is exclusion a sub-type of deprivation or is it a different concept with some overlapping dimensions?  For example were there people with higher levels of inclusion who also had high levels of deprivation, or are the two very highly correlated, in which case deprivation could be used as a proxy for exclusion.  Finally a short discussion on the potential generalisability of these findings for other countries would be welcome.  Were these measures specifically tailored to South Africa or are they standardised internationally? 

Author Response

We thank all three reviewers for their comments, and we believe the revisions we have made to the paper in light of these comments have improved its clarity and focus.

We have included a revised version with tracked changes of insertions to facilitate the review of the revised version.

Reviewer 3:

The article presents findings from a study which used data from the Individual Deprivation Measure for South Africa to measure inclusion of people with mild and moderate/severe disability.  Overall the findings indicated a number of variables in the dataset which could be used for this purpose, but also identified some gaps and limitations.  The article is well written and the methods and findings are clearly presented, and limitations are also extensively discussed.  I only have 2 suggestions for improvement.  Firstly a bit more information about the study should be provided, in particular overall numbers of respondents and numbers of people with mild and moderate disability who responded to the survey, and ideally the response rate as well.

As noted with respect to comments from Reviewers 1 and 2, additional information and clarity about the implementation of the survey and the analysis have been included in Section 2 (Materials and methods). These provide additional information (and greater clarity) around the detail of the implementation of the IDM South Africa study (Section 2.2), as well as information on the data analysis methods (including response rates) presented in the paper (Section 2.5).

Section 2.3 describes the categorisation of the disability severity measure and provides information about number of respondents in each category—there were 6,952 with no disability, 812 with mild disabilities and 888 with moderate or severe disabilities. Detailed information about the demographic characteristics of the individuals in each of these three groups is provided in Table 1 including the breakdown by gender, age category, population group and educational completion. Table 2 describes the onset of disability for those categorised as having mild and moderate disabilities, by gender and age group.

Secondly some more discussion on the relationship between inclusion/exclusion and deprivation would help contextualise these findings.  Is exclusion a sub-type of deprivation or is it a different concept with some overlapping dimensions? For example were there people with higher levels of inclusion who also had high levels of deprivation, or are the two very highly correlated, in which case deprivation could be used as a proxy for exclusion. 

With the revisions and refinements of the objectives of the paper, we have also revised the text in Sections 4 and 5 (the discussion and conclusions) to better reflect these. In the process, the few references that we had made to the links between inclusion/exclusion and deprivation have been removed (though the lessons drawn from using datasets designed for one purpose for an alternate purpose obviously remain). We have done this because the paper does not propose any specific relationship between inclusion/exclusion and deprivation, but does utilise an existing dataset specifically designed to measure levels of deprivation as an example of the way the proposed framework of inclusion measurement could be operationalised, and to examine the feasibility of doing so. The paper does seem to demonstrate a relationship between the two, because this work has demonstrated that it is feasible (within certain parameters) to use a dataset designed to measure deprivation to measure some indicators of important domains of inclusion. However, to include any substantive discussion of theoretical or empirical relationships between the two concepts would be beyond the scope of this paper, but indeed would be an interesting line of enquiry.  

Finally a short discussion on the potential generalisability of these findings for other countries would be welcome.  Were these measures specifically tailored to South Africa or are they standardised internationally? 

We have included text in Section 4 (Discussion) that addresses these comments. In particular, the discussion at the end of Section 4.1 regarding inference and generalisability, which states that the results of the analysis from South Africa are unlikely to be generalisable, because of its unusual status of being categorised as a middle-income country, but with extremely high levels of inequality and poverty. However, we also note in Section 4.3 that we believe that the method is generalisable (i.e. that of utilising other existing datasets to populate the disability inclusion framework, as appropriate), and that this analysis could be replicated using other IDM datasets, as the IDM survey tools were designed to be standardised across different contexts.

Round 2

Reviewer 1 Report

Great job on revisions.

Author Response

No response if required for Reviewer 1 as no revisions were requested.